# The Impact of Soil Erosion on Biodiversity Conservation in Isiala Ngwa North LGA, Southeastern Nigeria

**Godson Chinonyerem Asuoha, Uchenna Paulinus Okafor \*** **, Philip Ogbonnia Phil-Eze** **and Romanus Udegbunam Ayadiuno**

Department of Geography, University of Nigeria, Nsukka 410001, Nigeria;
chinonyerem.asuoha@unn.edu.ng (G.C.A.); philip.phil-eze@unn.edu.ng (P.O.P.-E.);
romanus.ayadiuno@unn.edu.ng (R.U.A.)
\* Correspondence: uchenna.okafor@unn.edu.ng; Tel.: +234 (0)8034390593

**Abstract:** The impact of soil erosion on the conservation of biodiversity in Isiala Ngwa North LGA, Southeastern Nigeria was examined. Data were obtained through focus group discussions and plant species enumeration. Diversity indices of plant species were derived from quadrat analysis using Shannon Wiener's diversity index. Eighteen soil samples were collected from agricultural erosion sites in the study area and analysed in the laboratory. The results obtained were analysed using principal component analysis (PCA). The rotated component matrix of the soil properties, as well as plant and animal diversity indices from the PCA isolated three components that together explained 93.821% of the observed variation. The results show that bush clearing in the form of slash and burn, uncoordinated bush burning and harvesting of plant species are the activities that cause soil erosion in the study area. Agro-forestry, bush fallowing, reforestation and legislation on indiscriminate harvesting of plant species were recommended.

**Keywords:** biodiversity conservation; environmental degradation; soil erosion; environmental sustainability; species diversity

## 1. Introduction

Land is a resource for agricultural activities. Unregulated increase in land-use causes soil erosion and loss of biodiversity. Erosion is widely recognized as one of the main threats to soil [1]. A challenge to the sustainability of agriculture in tropical regions is soil erosion. In addition, a critical global land degradation phenomenon affecting human beings is soil erosion. This is because humanity's basic sources of livelihood are obtained from the land. Feiznia and Nosrati [2],Chappell et al. [3],Mohawesh et al. [4] and the IPCC [5] are of the opinion that land use changes worldwide have been increasing the rate of soil erosion and loss of biodiversity. Intensive agricultural activities are reported to lead to soil erosion and loss of soil biodiversity [6–10].

Soil erosion results in the depletion of below-ground biodiversity, which includes soil microorganisms. Vallejo et al. [11] and Gardi et al. [12], quoted in Lui et al. [13], state that soils are one of the main living places of microorganisms and that microorganisms are involved in the decomposition of organic matter and help in the cycling and transformation of soil organic matter and soil nutrients, which include carbon, nitrogen, phosphorus, and sulphur. These soil nutrients enhance agricultural productivity if they are not degraded by soil erosion. Worldwide, soil is being degraded at a rapid rate. Globally, through soil erosion, about 2.8 tonnes of soil are lost per hectare annually [14]. The Centre for Science and Environment [15] states that about 25–30% of total cultivated

land in India is affected by soil erosion. Also, Le Roux et al. [16] state that, in South Africa, over 70% of the nation's land surface has been negatively impacted by varying levels and types of soil erosion. Similarly, the FAO [17], quoted in Henderson-Sellers [18],indicates that "without any conservation measures, the total area of rain-fed cropland in developing countries such as Latin America, Asia and Africa would decrease by 544 million hectares in the long term because of soil erosion and land degradation."Loss of soil from agricultural land may cause environmental impacts as well as reducing soil productivity. Lal et al. [19] state that soil fertility, organic matter in the soil, plant rooting depth and plant-available water reserves can be decreased by soil erosion. Kumar and Pani [20] state that "soil's physical degradation affects crop growth and yield by decreasing root depth, water availability and nutrient reserves. Thus, it leads to yield loss by affecting soil organic carbon, nitrogen, phosphorus, and potassium contents and soil pH."Scherr [21] stated that, "the effects of soil degradation vary with the initial soil conditions, types of soil, extent of soil degradation and crops." Increased food production is required by the future world population [22], which is said to have grown to 7.06 billion by the middle of 2012, after having crossed the 7 billion mark in 2011 [23]. Also, Engelman [24] highlighted that "the 79.3 million people added to the overall global population each year has been consistent for nearly a decade." There is a need, therefore, to increase agricultural production to feed these additional millions of people each year. Without good-quality and nutrient-rich soil, this is not possible. Hence, damage, through erosion or in any other form, to the soil indirectly damages agricultural production and, ultimately, food security. According to Wall et al. [25], "the implication of soil erosion extends beyond the removal of valuable topsoil. In fact, crop emergence, growth and yield are directly affected through the loss of soil natural nutrients."Bathrellos et al. [26], state that "the main on-site impact is the reduction of soil quality caused by the loss of the nutrient-rich upper layers of the soil and the reduced water-holding capacity of many eroded soils." The authors of [19] state that "The erosion of soil is one of several natural and human threats to sustained soil productivity, which may become irreversible if not mitigated." Tunji and Jeje [27] highlight that "soil erosion is aggravated by factors such farming system, soil management practices and rural poverty due to the pressure on soil." They also state that "erosion threatens man's source of food, livelihood and destroys man's property and investments."

Thus, this study focuses on the impact of soil erosion on biodiversity conservation in Isiala Ngwa LGA, Southeastern Nigeria. The vegetation type is rainforest, but has been largely degraded due to human activities including agriculture and agriculturally induced soil erosion. Igbozurike [28] states that "the vegetation is characterized by an abundance of plant species which sometimes exceed 150 different species per hectare, and its great diversity distinguishes it as one of the richest of all terrestrial ecological system in terms of biomass productivity."Unfortunately, there has been a high level of decimation of biodiversity, resulting in decreased biomass productivity. This development could be attributed to intensive agriculture, which contributes to soil erosion in the study area. Dominant plant species include *Khaya ivorensis*, *Melicia excela*, *Pentaclethra macrophylla*, *Elaeis guinensis*, and *Raphia vinifera*. Wildlife species include *Python sabae*, *Hyena stirata*, *Protoxerus strangei*, *Philantomba maxwelli*, *Cricetomys gambianus*, etc. The population is largely rural and numbered 154,083 in the 2006 census [29].

## 2. Material and Methods

The study was based on Isiala Ngwa North LGA of Abia State Southeastern, Nigeria. It is an LGA with 40 communities covering an area of about 83.5km$^2$ (Figure 1) and lies approximately within latitudes 05°21' and 05°29' N and longitudes 07°18' and 07°22' E. It is within the Af climate of Koppen's classification, with two distinct seasons, the rainy and dry seasons. The rainy season is between April and October, while the dry season is from November to March.

The impact of soil erosion on biodiversity was analysed. This was done using data from the biodiversity inventory obtained from the sites of the dominant agricultural land use practices. Biodiversity indices were calculated from the sites of five major agricultural land use practices, while soil samples were collected from the sampled quadrats for analysis. The calculation of biodiversity indices for the species in the area was done using the formula after Hill [30].

$$\text{Biodiversity Index} = \frac{the\ number\ of\ species\ in\ the\ area}{the\ number\ of\ individuals\ in\ the\ area} \tag{1}$$

Principal component analysis was used to analyse the relationship between agricultural land use practices and biodiversity. Focus group discussion was used to obtain data from farmers and hunters in the area. The relationship between soil and biodiversity in the area is presented in Table 1.

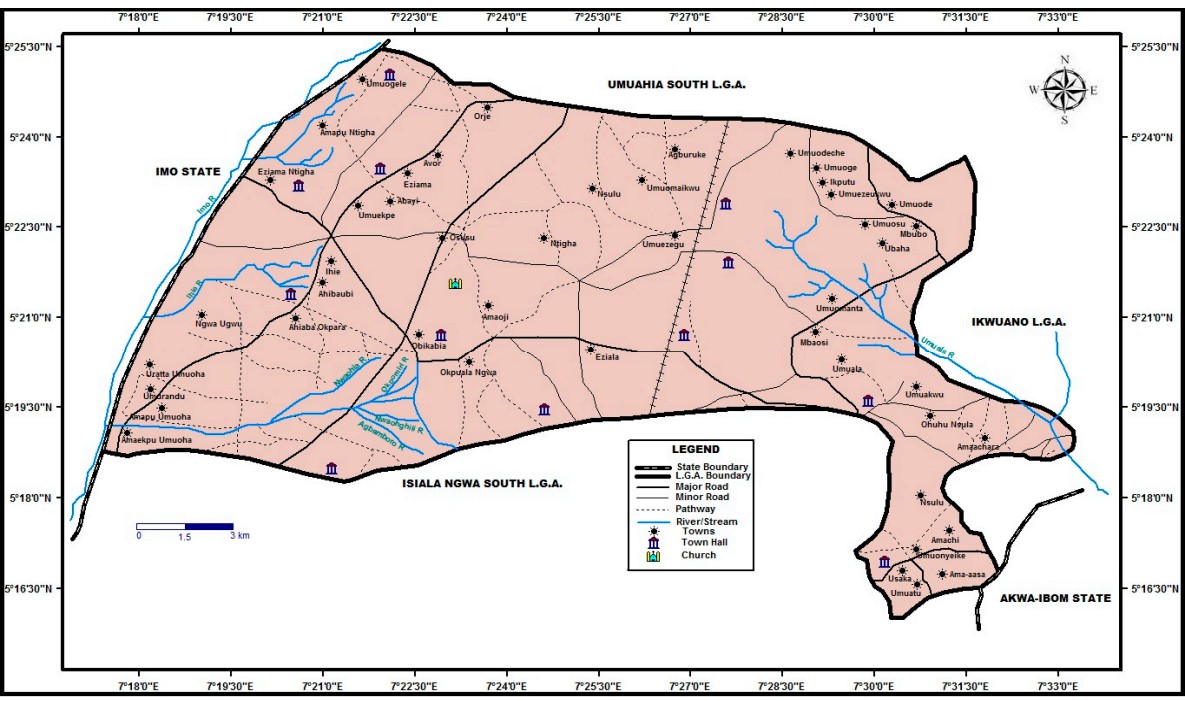

**Figure 1.** Isiala Ngwa North LGA, showing the communities.

**Table 1.** Soil characteristics and biodiversity in the area.

| S/N | Sample | % Sand | % Silt | % Clay | pH Soil | P mg/kg | % N | % OC | % OM | % Ca | Mg Cmol | % Na | % Ex.A1 | Plants | Animals | Biodiversity |
|---|---|---|---|---|---|---|---|---|---|---|---|---|---|---|---|---|
| 1 | Intercropping | 81.60 | 6.20 | 12.00 | 4.10 | 13.30 | 0.126 | 1.147 | 1.977 | 3.20 | 1.60 | 1.973 | 0.68 | 2.73 | 1.85 | IC=0.089 |
| 2 | Mixed farming | 79.80 | 9.70 | 10.33 | 4.a27 | 13.90 | 0.154 | 1.437 | 2.477 | 3.00 | 2.00 | 2.773 | 1.00 | 2.82 | 1.67 | MF=0.059 |
| 3 | Plantation agriculture | 73.80 | 18.20 | 8.00 | 4.57 | 23.80 | 0.105 | 0.880 | 1.700 | 5.00 | 2.20 | 0.480 | 0.12 | 2.78 | 1.66 | PA=0.071 |
| 4 | Bush fallowing | 78.80 | 12.20 | 9.00 | 4.30 | 19.50 | 0.088 | 0.717 | 1.230 | 4.13 | 2.20 | 1.280 | 0.80 | 2.75 | 1.67 | B.F=0.058 |
| 5 | Animal husbandry | 74.80 | 13.70 | 11.50 | 4.30 | 16.10 | 0.126 | 1.167 | 2.010 | 4.40 | 1.40 | 0.760 | 0.66 | 2.81 | 1.60 | A.H=0.053 |

Source: Field work, 2018.

## 3. Results and Discussion

From the result, a soil/biodiversity relationship table was generated as shown in Table 1. Data on physical and chemical properties of soil in the area are given in Appendix A; correlation result between soil and biodiversity are presented in Appendix B. The raw data are also given as Appendix A.

The Spearman's rank correlation coefficient was run to generate correlation matrix for the soil properties and plant diversity index, animal diversity index, plant and animal diversity index and biodiversity index (Appendix B).

The equation for Spearman rank correlation is:

$$\text{rR} \; = \; 1 \; - \; \frac{6 \sum i d i 2}{n(n2-1)}. \tag{2}$$

In the correlation between soil properties and plant diversity index, there is a correlation between soil and plant species diversity index. Hence, there is a correlation between plant index and sand, as well as plant index and clay, although the coefficient is negative. The negative coefficient between plant index and sand implies that, where sand is high, there will be less plant species diversity. That of plant index and clay means that where the clay content of the soil is high, it would adversely affect plant growth. The reason is that the water may not penetrate the clay and reach the roots of certain plant species. It is only those plants whose roots are within the clayey part that would thrive. There is a positive correlation between plant index and silt; the same applies to plant index and soil pH.

The positive correlation between plant diversity index and silt means that high silt content favours plant growth. On the other hand, when the soil pH is high, there would be less plant diversity. In the correlation between soil properties and animal diversity index, it could be seen that there is a relationship between animal index and sand (0.671), animal index and silt (-0.671), though with a negative correlation coefficient. This means that where sand content is high, there would be more animal diversity, and vice versa. For animal index and silt, where there is high silt content, there would be low animal diversity. The same also applies to animal index and soil pH (-0.574). This indicates that less soil pH entails high animal diversity. When the soil pH is high, there would be less plant diversity, which in turn negatively affects wildlife.

For the correlation among soil properties, plant and animal diversity indices, there is also correlation between soil and plant and animal diversity indices. The implication is that the more the soil properties are in the right proportions, the higher the diversity of plant and animal species. Hence, the coefficient for plant index and sand is -0.564. This means that more sand content in the soil implies less plant diversity. The reason is that all the soil properties must be in the right proportions for plants to grow well. That of plant index and silt is 0.564, while the coefficient for plant index and clay is -0.616. This means that, if the silt content is high, it will not favour plant growth. Similarly, as clay retains moisture, smaller plants whose roots do not penetrate beyond the level of clay may not do well. It is only those with stronger roots that penetrate beyond this level that would flourish. The correlation coefficient between plant index and soil pH is 0.526. As for that between plant diversity index and soil pH, the higher the soil pH, the less diverse the plant species in an area. On the other hand, the correlation coefficient between animal index and sand is 0.671. This, however, implies that the higher the sand content, the lower the animal diversity. That of animal index and silt is -0.671. This implies that the lower the silt, the higher the animal diversity. While the correlation coefficient between animal index and soil pH is -0.574, that between animal index and nitrogen is 0.000. This implies that the higher the soil pH, the lower the animal diversity in an area. There is no relationship between animal diversity index and nitrogen.

Surprisingly, in the correlation between soil properties and biodiversity, there was no relationship between sand and biodiversity. The same applies to silt and biodiversity. However, there was a weak correlation between clay and biodiversity. The same also applies to exchangeable aluminium (Ex.Al) and biodiversity. There was also a moderate correlation between Mg and biodiversity, but it was not

encouraging. However, the results of the correlation between soil properties and biodiversity is a general reflection of the effect of human activities on soil. This, in turn, shows the extent to which these activities have aggravated soil erosion, which has also discouraged biodiversity conservation. Generally, eroded soil can barely support robust vegetation and so cannot harbour wildlife. Furthermore, to ascertain the underlying factors responsible for the observed relationship between the soil properties and biodiversity, the soil test result was subjected to PCA. The first analysis was run between soil properties and the plant index. The second was between soil and the animal index. The third was run between soil and the plant and animal indexes, while the fourth was between soil and the biodiversity index. The PCA of the soil and plant index is shown in Table 2.

**Table 2.** Rotated component matrix of the soil properties and plant diversity index in the study area.

|  |  | Component | | |
|---|---|---|---|---|
|  | **Variables** | **1** | **2** | **3** |
| X1 | % of sand | −0.984 * | −0.027 | −0.097 |
| X2 | % of silt | 0.904 * | −0.084 | 0.404 |
| X3 | % of clay | −0.413 | 0.252 | −0.875 * |
| X4 | % of soil pH | 0.807 * | 0.006 | 0.589 |
| X5 | % of potassium | 0.704 * | −0.418 | 0.569 |
| X6 | % of nitrogen | −0.297 | 0.919 * | -0.242 |
| X7 | % Organic carbon (OC) | −0.299 | 0.903 * | −0.305 |
| X8 | % Organic matter (OM) | −0.187 | 0.941 * | −0.237 |
| X9 | % of Ca | 0.907 * | −0.385 | 0.158 |
| X10 | % of Mg | −0.010 | −0.218 | 0.974 * |
| X11 | % of Na | −0.868 * | 488 | 0.082 |
| X12 | % of Ex. Al | −0.788 * | 0.277 | −0.117 |
|  | Plant index | 0.618 | 0.705 | 0.300 |
|  | Eigenvalue | 5.893 | 3.800 | 2.902 |
|  | % of variance | 45.333 | 29.233 | 22.322 |
|  | Cumulative % | 45.333 | 74.566 | 96.888 |

* significant loadings ≥ +/ − 0.70.

The results of the rotated component matrix above show that three components were extracted from the 12 variables. Component 1 has significant loadings on seven variables. The variables with negative signs are $X_I$ (% of sand), $X_{II}$ (% of Na) and $X_{12}$ (Ex. Al). This means that the more agricultural land use practices adversely affect these soil properties, the lower the soil quality following the attendant soil erosion. This, in turn, implies that such soil would not support more plant diversity index. The variables with positive signs are $X_2$ (% of silt), $X_4$ (% of soil pH), $X_5$ (% of potassium), and $X_9$ (% of Ca). This means that as long as these practices do not have adverse effects on the soil, there is bound to be more plant species diversity. The underlying factor becomes the effect of farming systems on soil physicochemical properties. The component has an eigen value of 5.893 and explains 45.333% of the total variance.

Component II has significant loadings on three variables viz $X_6$ (% of nitrogen), $X_7$ (% of Organic Carbon (%OC)), and $X_8$ (% of Organic Matter (%OM)). The heavy loadings on these variables denote that if the land use practices do not impact negatively on nitrogen, organic carbon and organic matter, there would be a higher diversity of plant species and vice versa. This is because these properties of the soil favour plant growth in an area. In other words, the plant diversity index (705*) is in agreement with variables $X_6$, $X_7$ and $X_8$. The underlying factor here is the carbon–nitrogen ratio. The component has an eigen value of 3.800 and explains 29.233% of the total variance.

Component III loads heavily on two variables. They are $X_3$ (-0.875) and $X_{10}$ (0.974). Variable $X_3$ has a negative loading (-0.875), although high, which means that the more the soil lacks clay, the more it supports plant species diversity. This is because clay retains water and does not allow it to permeate. This adversely affects plant diversity. Variable X10, which loads with a positive sign, implies that as long as the quantity of magnesium is not adversely affected by the farming practices, there is bound to

be plant diversity in such an area. Hence, the component has an eigen value of 2.902 and contributes 22.322% of the total variance. The underlying factor here becomes the index of soil fertility.

PCA was used to ascertain the major factors responsible for the observed variation in the correlation between soil properties and animal diversity index. The rotated component matrix is presented in Table 3.

**Table 3.** Rotated component matrix of the soil properties and animal diversity index in the study area.

|  | | Component | | |
|---|---|---|---|---|
|  | Variables | I | II | III |
| X1 | % of sand | 0.991 * | 0.124 | −0.028 |
| X2 | % of silt | −0.913 * | −0.227 | 0.339 |
| X3 | % of clay | 0.425 | 0.330 | −0.842 * |
| X4 | % of soil pH | −0.826 * | −0.141 | 0.527 |
| X5 | % of potassium | −0.654 | −0.543 | 0.513 |
| X6 | % of nitrogen | −0.175 | 0.946 * | −0.222 |
| X7 | % of OC | 0.180 | 0.934 * | −0.284 |
| X8 | % of OM | 0.071 | 0.942 * | −0.227 |
| X9 | % of Ca | −0.846 * | −0.525 | 0.090 |
| X10 | % of Mg | −0.018 | −0.232 | 0.971 * |
| X11 | % of Na | 0.772 * | 0.618 * | 0.145 |
| X12 | % of Ex. Al | 0.673 * | 0.458 | −0.044 |
|  | Animal diversity index | 0.723 | −0.208 | −0.203 |
|  | Eigenvalue | 5.463 | 4.111 | 2.561 |
|  | % of variance | 42.024 | 31.625 | 19.698 |
|  | Cumulative % | 42.024 | 73.650 | 93.347 |

\* significant loadings ≥ +/− 0.70.

Table 3 shows three components. Component I has significant loadings on six variables. Variables $X_I$ (% of sand), and $X_{II}$ (% of Na) have positive signs and a high load. This means that if the agricultural land-use practices have a small negative impact on these soil properties, the soil would support plant growth, which in turn encourages animal diversity in the area. Variables $X_2$ (% of silt, $X_4$ (% of soil pH), and $X_9$ (% of Ca) have highly negative loadings, meaning that animal diversity would be discouraged if such soil properties are adversely impacted due to soil erosion. The component has an eigenvalue of 5.463 and explains 42.024% of the total variance. The underlying factor is the general disposition of soil properties towards animal species diversity.

For component II, there are significant loadings on variable X6(% of nitrogen), X7(% of OC), X8(% of OM) and X11(% of Na). The significantly positive loadings here imply that these soil properties denote soil fertility in an area and as such would encourage plant growth. This plant growth would aid animal diversity, especially for those larger animals that cannot hide under grasses. Even those that burrow in the soil are favoured. So, any farming systems that favour these soil properties in turn favour animal species diversity. The component has an eigen value of 4.111 and explains 31.625% of the total variance. The underlying factor here is favourable habitat.

Component III has heavy loadings on two variables, viz: X3(% of clay) and X10(% of Mg). Variable X3 has a highly negative loading, meaning that when the percentage of clay is negatively impacted by land use practices, the life of certain animals is endangered there. This is even more pronounced in the case of burrowing animals like *Rattus rattus.* Variable X10 has a high positive loading, meaning that when the percentage of magnesium in the soil diminishes due to farming systems, animal diversity is limited. The component has an eigen value of 2.561 and explains 19.698% of the total variance. The underlying factor here is clay mineral impact. The three components together explain 93.347% of the observed variation, leaving the remaining 6.653% unexplained.

The rotated component matrix of the soil properties with plant diversity index and animal diversity index is shown in Table 4.

**Table 4.** Component matrix of the soil properties, plant and animal indices in the study area.

| | Variables | I | II | III |
|---|---|---|---|---|
| | | **Component** | | |
| X1 | % of sand | −0.983 * | 0.180 | −0.004 |
| X2 | % of silt | 0.908 * | −0.270 | 0.319 |
| X3 | % of clay | −0.430 | 0.332 | −0.838 * |
| X4 | % of soil pH | 0.831 * | −0.176 | 0.506 |
| X5 | % of potassium | 0.637 | −0.568 | 0.509 |
| X6 | % of nitrogen | −0.128 | 0.948 * | −0.244 |
| X7 | % of OC | −0.136 | 0.935 * | −0.305 |
| X8 | % of OM | −0.026 | 0.937 * | −0.252 |
| X9 | % of Ca | 0.818 * | −0.570 | 0.083 |
| X10 | % of Mg | 0.033 | −0.208 | 0.976 * |
| X11 | % of Na | −0.732 * | 0.665 | 0.149 |
| X12 | % of Ex. Al | −0.644 | 0.499 | −0.035 |
| | Plant index | 0.796 | 0.564 | 0.220 |
| | Animal Index | −0.742 | −0.176 | −0.179 |
| | Eigenvalue | 5.911 | 4.626 | 2.597 |
| | % of variance | 42.223 | 33.045 | 18.553 |
| | Cumulative | 42.223 | 75.268 | 93.821 |

* significant loadings ≥ $+/-0.70$.

Table 4 shows that there are three components resulting from the 12 variables. Component 1 has significant loadings on five variables. Variables $X_I$ (% of sand) and $X_{II}$ (% of Na) have highly negative loadings. This means that, as these soil components are impacted negatively by the farming system in the area, animal diversity tends to be suppressed. Variables $X_2$ (% of silt), $X_4$ (% of soil pH) and $X_9$ (% of Ca) have significant loadings, meaning that the more these soil properties are moderately impacted by agricultural land-use practices, the higher the plant diversity index. The component has an eigen value of 5.911 and explains 42.223% of the total variance. The underlying factor here is soil conditions for species diversity.

Component II has three variables with significant loadings. They are $X_6$ (% of nitrogen), $X_7$ (% of OC)) and $X_8$ (% of OM)). This implies that as long as nitrogen, organic carbon and organic matter are favoured by the farming systems in an area, plant species would flourish. Hence, these are soil nutrients that enhance plant growth. The component has an eigen value of 4.626 and represents 33.045% of the total variance. The underlying dimension is favourable conditions for plant species diversity. Component III has significant loadings on two variables viz: $X_3$ (% of clay) and $X_{10}$ (% of Mg). For variable $X_3$, the higher the percentage of clay, the lower the animal species diversity, whereas the higher the magnesium content in the soil the higher the plant species diversity. The component has an eigen value of 2.597 and explains 18.553% of the total variance. The underlying factor here is the impact of soil minerals on biodiversity. However, the three components jointly explain 93.821% of the variation on the input data, leaving 6.1792 unexplained. The PCA on soil properties and biodiversity index is shown in Table 5.

**Table 5.** Component matrix of soil properties and biodiversity index in the study area.

|  | | Component | | |
|---|---|---|---|---|
| | **Variables** | **I** | **II** | **III** |
| X1 | % of sand | −0.961 * | −0.078 | −0.028 |
| X2 | % of silt | 0.889 * | −0.189 | 0.338 |
| X3 | % of clay | −0.423 | 0.316 | −0.842 * |
| X4 | % of soil pH | 0.823 * | −0.074 | 0.549 |
| X5 | % of potassium | 0.701 * | −0.474 | 0.532 |
| X6 | % of nitrogen | −0.240 | 0.950 * | −0.199 |
| X7 | % of OC | −0.249 | 0.930 * | −0.266 |
| X8 | % of OM | −0.121 | 0.974 * | −0.190 |
| X9 | % of Ca | 0.874 * | −0.465 | 0.096 |
| X10 | % of Mg | 0.015 | −0.247 | 0.968 * |
| X11 | % of Na | −0.825 * | 0.547 | 0.137 |
| X12 | % of Ex. Al | −0.835 * | 0.240 | −0.146 |
| | Biodiversity index | 0.759 | −0.038 | 0.494 |
| | Eigenvalue | 5.915 | 3.723 | 2.786 |
| | % of variance | 45.504 | 28.640 | 21.428 |
| | Cumulative % | 45.504 | 74.144 | 95.572 |

* Significant loadings $\geq +/-0.70$.

Table 5 shows that three components were extracted from the 12 variables. Component I has significant loadings on seven variables. The variables with negative signs are $X_I$ (% of sand), $X_{II}$ (% of Na) and $X_{12}$ (% of Ex. Al), meaning that the more negatively these soil properties are impacted, the lower the biodiversity index. For the four variables $X_2$ (% of silt), $X_4$ (% of soil pH), $X_5$ (% of potassium) and $X_9$ (% of Ca), the heavy loadings imply that an increase in these soil properties increases biodiversity index. The underlying component here is conditions for biodiversity enhancement. The component has an eigen value of 5.915 and accounts for 45.504% of the total variance.

In the case of component II, only three variables are significantly loaded. They are $X_6$ (% of Nitrogen), $X_7$ (% of OC) and $X_8$ (% of OM). Their high loadings imply that the more pronounced these properties are in the soil, the higher the biodiversity index. This is because these three elements favour plant growth, which in turn encourages biodiversity. The component has an eigen value of 3.723 and explains 28.640% of the total variance. The underlying dimension is the impact of favourable soil properties on biodiversity.

Component III has two variables with significant loadings. They are $X_3$ (% of clay) and $X_{10}$ (% of Mg). This means that while high clay content in the soil impacts negatively on the biodiversity index, high Mg content impacts positively on biodiversity. In terms of X10, more Mg in the soil attracts more biodiversity. This is because Mg is an essential nutrient for plant growth. The component has an eigen value of 2.786 and accounts for 21.428% of the total variance. The underlying factor here is effect of clay/Mg relationship on biodiversity. The three components therefore account for 95.572% of the observed variation, leaving 4.428% unexplained.

A summary of the focus group discussion with some farmers and hunters in the area is presented in Table 6.

**Table 6.** Results of the key informant interviews and focus group discussions with farmers in the area.

| Questions Raised | Responses from Respondents | Researcher's Comments |
|---|---|---|
| Concerning the farming systems practiced | Farming system is mainly peasant or subsistence farming. The major practice is bush fallowing, with a fallow period of 2–3 years | Short fallow periods do not favour biodiversity conservation. |
| About farm tools used and type of crops planted | Farm tools are hoes, knives and spades. The type of crops planted are cassava, yam, garden eggs and vegetables, with the yield determined by soil fertility. | Farming systems determine the tools used. Hence simple farm tools used for subsistence farming. |
| What type of animals are kept in the area | They include ruminants, e.g., goats, sheep and cattle. Some livestock roam about (free range) while others are semi-free range—partially housed and sometimes left on their own. | Leaving the livestock to roam about exposes the soil to erosion and eventual biodiversity depletion. |
| Concerning the effect of constant bush clearing on biodiversity | They agreed that it has negative impacts on wildlife and plant species in the area | Most mammals that require forest areas for habitation are now absent. Hence, this practice causes loss of habitat due to soil erosion. |
| As to whether there are protected areas in the area | There are informal protected areas, e.g., waste land around shrines, along ancestral bush tracks. | More of these protected areas are advocated. |
| Concerning sustainable agricultural production and biodiversity conservation | They suggested controlled burning, i.e., gathering the grasses together and burning them. So, organic farming is advised. | If organic farming is practiced, biodiversity conservation is assured. |
| In terms of the use of inorganic fertilizer | It was obtainable in the past due to long fallow periods. However, population growth has resulted in land scarcity, leading to short fallow periods. | This does not aid biodiversity recuperation. |
| About continuous cropping | This is mostly done at the back of people's houses, schools, farms, etc. | Over time, this may lead to perpetual loss of soil fertility. |
| About hunting and bush burning in relation to biodiversity | Some hunters set the bush ablaze so as to catch some animals. Some widows do the same in order to clear their land for farming as they have nobody to help them. | Bush burning is very detrimental to wildlife and vegetation and should be discouraged. It also leads to erosion. |

Source: Field work, 2018.

## 4. Conclusions

Intensive agricultural activities are reported to lead to a loss of soil biodiversity and are cited as a source of environmental degradation [31]. The study has revealed both the general and specific effects of human activities on the soil. It has also shown how various agricultural land use practices have impacted the soil, culminating in soil erosion. Also, the effects of the activities of some hunters in the area trickled down to affect the intensity of soil erosion in the area. This, in turn, was evidenced in the level of biodiversity loss, as was indicated by the low biodiversity indices in some sampled quadrats. The soil analyses and the relationship between soil physicochemical properties and biodiversity have shown the gravity of soil erosion and its attendant impact on biodiversity conservation or depletion. In the light of the above, therefore, we recommend bush fallowing with long fallow periods,

reforestation in areas that have been cleared by soil erosion, and constant application of manure to regenerate reforested soil. There should also be legislation on the arbitrary harvesting of plant and animal species. Rural communities should also be discouraged from indiscriminate bush burning. With the implementation of the recommendations highlighted above, the achievement of sustainable development goals (SDGs) can become be a reality.

**Author Contributions:** Ideas; formulation or evolution of overarching research goals and aims. Development or design of methodology; creation of models, G.C.A.; Application of statistical, mathematical, computational, or other formal techniques to analyze or synthesize study data. Preparation, creation and/or presentation of the published work, specifically writing the initial draft (including substantive translation). U.P.O.; Preparation, creation and/or presentation of the published work by those from the original research group, specifically critical review, commentary or revision – including pre- or post-publication stages. Oversight and leadership responsibility for the research activity planning and execution, including mentorship external to the core team. P.O.P.-E.; Acquisition of the financial support for the project leading to this publication. Provision of study materials, reagents, materials, patients, laboratory samples, animals, instrumentation, computing resources, or other analysis tools. R.U.A.; Programming, software development; designing computer programs; implementation of the computer code and supporting algorithms; testing of existing code components. R.U.A.; Management activities to annotate (produce metadata), scrub data and maintain research data (including software code, where it is necessary for interpreting the data itself) for initial use and later re-use. U.P.O.

**Funding:** This research received no external funding.

**Conflicts of Interest:** The authors declare no conflict of interest.

# Appendix A

**Table A1.** Physical and chemical properties of soils under respective agricultural land use practices.

| S/N | Sample | % Sand | % Silt | % Clay | TEXT | PH H$_2$O | P mg/kg | % N | % Oc | % OM | Ca | MgCmol | Na | Ex.A1 |
|---|---|---|---|---|---|---|---|---|---|---|---|---|---|---|
| 1 | Uncultivated (control I) | 78.8 | 12.20 | 8.0 | LS | 4.0 | 26.2 | 0.168 | 1.74 | 3.00 | 4.80 | 1.20 | 8.04 | 1.20 |
| 2 | Uncultivated (control II) | 83.8 | 8.2 | 8.0 | LS | 4.3 | 24.7 | 0.154 | 1.67 | 2.88 | 9.20 | 4.80 | 0.40 | 0.00 |
| 3 | Uncultivated (control III) | 81.3 | 10.2 | 8.0 | | 4.15 | 25.45 | 0.161 | 1.71 | 2.94 | 7.00 | 3.00 | 4.22 | 0.00 |
| 4 | Crop farming I | 83.8 | 4.2 | 12.0 | LS | 4.2 | 13.9 | 0.140 | 1.20 | 2.07 | 4.40 | 2.80 | 1.36 | 0.48 |
| 5 | Crop farming II | 79.8 | 8.2 | 12.0 | LS | 4.0 | 12.6 | 0.112 | 1.09 | 1.88 | 2.00 | 0.40 | 1.60 | 0.88 |
| 6 | Crop farming III | 81.2 | 6.2 | 12.0 | | 4.1 | 13.25 | 0.126 | 1.15 | 1.98 | 3.20 | 1.60 | 2.96 | 0.68 |
| 7 | Mixed farming I | 79.8 | 10.2 | 10.0 | LS | 4.2 | 12.4 | 0.182 | 1.67 | 2.88 | 3.20 | 2.80 | 1.92 | 0.64 |
| 8 | Mixed farming II | 79.8 | 9.2 | 11.0 | LS | 4.3 | 15.4 | 0.126 | 1.20 | 2.07 | 2.80 | 1.20 | 2.24 | 1.36 |
| 9 | Mixed farming III | 79.8 | 9.7 | 10.0 | | 4.3 | 13.9 | 0.154 | 1.44 | 2.48 | 3.00 | 2.00 | 4.16 | 1.00 |
| 10 | Plantation agriculture I | 73.8 | 18.2 | 8.0 | SL | 4.6 | 20.0 | 0.126 | 1.16 | 2.00 | 4.00 | 1.60 | 0.64 | 0.08 |
| 11 | Plantation agriculture II | 73.8 | 18.2 | 8.0 | SL | 4.5 | 27.6 | 0.084 | 0.81 | 1.40 | 6.00 | 2.80 | 0.32 | 0.16 |
| 12 | Plantation agriculture III | 73.8 | 18.2 | 8.0 | | 4.6 | 23.8 | 0.105 | 0.67 | 1.70 | 5.00 | 2.20 | 0.48 | 0.12 |
| 13 | Bush fallowing I | 79.8 | 12.2 | 8.0 | SL | 4.5 | 20.7 | 0.077 | 0.93 | 1.60 | 5.00 | 2.40 | 0.56 | 0.48 |
| 14 | Bush fallowing II | 77.8 | 12.2 | 10.0 | SL | 4.1 | 18.3 | 0.098 | 0.50 | 0.86 | 3.20 | 2.00 | 2.00 | 1.12 |
| 15 | Bush fallowing III | 78.8 | 12.2 | 9.0 | | 4.3 | 19.5 | 0.088 | 0.72 | 1.23 | 4.20 | 2.20 | 1.28 | 0.80 |
| 16 | Animal husbandry I | 79.8 | 12.2 | 8.0 | Ls | 4.4 | 15.4 | 0.112 | 1.01 | 1.74 | 4.80 | 1.60 | 0.64 | 0.43 |
| 17 | Animal husbandry II | 69.8 | 15.2 | 15.0 | SL | 4.1 | 16.8 | 0.140 | 1.32 | 2.28 | 4.00 | 1.20 | 0.88 | 0.88 |
| 18 | Animal husbandry III | 74.8 | 13.7 | 11.5 | | 4.3 | 16.1 | 0.126 | 1.17 | 2.01 | 4.40 | 1.40 | 0.76 | 0.66 |

**Appendix B**

The equation for Spearman rank correlation is:

$$rR = 1 - \frac{6 \sum_i d_i^2}{n(n^2 - 1)}, \tag{2}$$

where *n* is the number of data points of the two variables and $d_i$ is the difference in the ranks of the *i*th element of each random variable considered. The Spearman correlation coefficient, ρ, can have values from +1 to -1. A ρ of +1 indicates a perfect association of ranks, a ρ of zero indicates no association between ranks and a ρ of -1 indicates a perfect negative association of ranks. The closer ρ is to 0, the weaker the association between the ranks.

**Table A2.** Correlation matrix for soil properties and animal diversity indices.

|  | % Sand | % Silt | % Clay | %pH | % Pot | % Nit | % OC | % OM | % Ca | % Mg | % Na | % Ex.AI | Animal Index |
|---|---|---|---|---|---|---|---|---|---|---|---|---|---|
| % Sand | 1.000 |  |  |  |  |  |  |  |  |  |  |  |  |
| % Silt | −1.00 | 1.000 |  |  |  |  |  |  |  |  |  |  |  |
| % Clay | 0.700 | −0.700 | 1.000 |  |  |  |  |  |  |  |  |  |  |
| % P$^H$ | −0.975 | 0.975 | −0.821 | 1.000 |  |  |  |  |  |  |  |  |  |
| % Pot | −0.900 | −0.900 | −0.900 | 0.975 | 1.000 |  |  |  |  |  |  |  |  |
| % Nit | 0.462 | 0.462 | 0.564 | −0.605 | −0.718 | 1.000 |  |  |  |  |  |  |  |
| % Oc | 0.300 | 0.300 | 0.500 | −0.462 | −0.600 | 0.975 | 1.000 |  |  |  |  |  |  |
| % OM | 0.300 | 0.300 | 0.500 | −0.462 | −0.600 | 0.975 | 1.000 | 1.000 |  |  |  |  |  |
| % Ca | −0.900 | −0.900 | −0.500 | 0.872 | 0.800 | −0.616 | −0.500 | −0.500 | 1.000 |  |  |  |  |
| % Mg | −0.308 | −0.308 | −0.872 | 0.500 | 0.667 | −0.632 | −0.667 | −0.667 | 0.205 | 1.000 |  |  |  |
| % Na | 0.900 | 0.900 | 0.500 | −0.872 | −0.800 | 0.616 | 0.500 | 0.500 | −1.000 | −0.205 | 1.000 |  |  |
| % Ex.AI | 0.700 | 0.700 | 0.200 | −0.616 | −0.500 | 0.359 | 0.300 | 0.300 | −0.900 | 0.051 | 0.900 | 1.000 |  |
| Animal index | 0.671 | 0.671 | 0.224 | −0.574 | −0.447 | 0.000 | −0.224 | −0.224 | −0.447 | 0.229 | 0.447 | 0.224 | 1.000 |

**Table A3.** Correlation matrix for soil properties and plant diversity indices.

|  | % Sand | % Silt | % Clay | %pH | % Pot | % Nit | % Oc | % OM | % Ca | % Mg | % Na | % Ex.AI | Plant index |
|---|---|---|---|---|---|---|---|---|---|---|---|---|---|
| % Sand | 1.000 |  |  |  |  |  |  |  |  |  |  |  |  |
| % Silt | −1.000 | 1.000 |  |  |  |  |  |  |  |  |  |  |  |
| % Clay | 0.700 | −0.700 | 1.000 |  |  |  |  |  |  |  |  |  |  |
| % P$^H$ | −0.975 | 0.975 | −0.821 | 1.000 |  |  |  |  |  |  |  |  |  |
| % Pot | −0.900 | 0.900 | −0.900 | 0.975 | 1.000 |  |  |  |  |  |  |  |  |
| % Nit | 0.462 | −0.462 | 0.564 | −0.605 | −0.718 | 1.000 |  |  |  |  |  |  |  |
| % Oc | 0.300 | −0.300 | 0.500 | −0.462 | −0.600 | 0.975 | 1.000 |  |  |  |  |  |  |
| % OM | 0.300 | −0.300 | 0.500 | −0.462 | −0.600 | 0.975 | 1.000 | 1.000 |  |  |  |  |  |
| % Ca | −0.900 | 0.900 | −0.500 | 0.872 | 0.800 | −0.616 | −0.500 | −0.500 | 1.000 |  |  |  |  |
| % Mg | −0.308 | 0.308 | −0.872 | 0.500 | 0.667 | −0.632 | −0.667 | −0.667 | 0.205 | 1.000 |  |  |  |
| % Na | 0.900 | −0.900 | 0.500 | −0.872 | −0.800 | 0.616 | 0.500 | 0.500 | −1.000 | −0.205 | 1.000 |  |  |
| % Ex.AI | 0.700 | −0.700 | 0.200 | −0.616 | −0.500 | 0.359 | 0.300 | 0.300 | −0.900 | 0.051 | 0.037 | 1.000 |  |
| Plant index | −0.564 | 0.564 | −0.616 | −0.526 | −0.462 | 0.263 | −0.359 | 0.359 | −0.205 | 0.289 | −0.205 | 0.103 | 1.000 |

**Table A4.** Correlation matrix for soil properties and plant and animal diversity indices.

| | % Sand | % Silt | % Clay | %pH | % Pot | % Nit | % Oc | % OM | % Ca | % Mg | % Na | % Ex.Al | Plant index | Animal index |
|---|---|---|---|---|---|---|---|---|---|---|---|---|---|---|
| % Sand | 1.000 | | | | | | | | | | | | | |
| % Silt | −1.000 | 1.00 | | | | | | | | | | | | |
| % Clay | 0.700 | −0.700 | 1.000 | | | | | | | | | | | |
| % P$^H$ | −0.975 | 0.975 | −0.821 | 1.000 | | | | | | | | | | |
| % Pot | −0.900 | −0.900 | −0.900 | 0.975 | 1.000 | | | | | | | | | |
| % Nit | 0.462 | 0.462 | 0.564 | −0.605 | −0.718 | 1.000 | | | | | | | | |
| % Oc | 0.300 | −0.300 | 0.500 | −0.462 | −0.600 | 0.975 | 1.000 | | | | | | | |
| % OM | 0.300 | −0.300 | 0.500 | −0.462 | −0.600 | 0.975 | 1.000 | 1.000 | | | | | | |
| % Ca | −0.900 | 0.900 | −0.500 | 0.872 | 0.800 | −0.616 | −0.500 | −0.500 | 1.000 | | | | | |
| % Mg | −0.308 | −0.308 | −0.872 | 0.500 | 0.667 | −0.632 | −0.667 | −0.667 | 0.205 | 1.000 | | | | |
| % Na | 0.900 | 0.900 | 0.500 | −0.872 | −0.800 | 0.616 | 0.500 | 0.500 | −1.000 | −0.205 | 1.000 | | | |
| % Ex.Al | 0.700 | −0.700 | 0.200 | −0.616 | −0.500 | 0.359 | 0.300 | 0.300 | −0.900 | 0.051 | 0.900 | 1.000 | | |
| Plant index | −0.564 | 0.564 | −0.616 | −0.526 | 0.462 | 0.263 | 0.359 | 0.359 | 0.205 | 0.289 | −0.205 | −0.103 | 1.000 | |
| Animal index | 0.671 | −0.671 | 0.224 | −0.574 | −0.447 | 0.000 | −0.224 | −0.224 | −0.447 | 0.229 | 0.447 | 0.224 | −0.459 | 1.000 |

**Table A5.** Correlation matrix for soil properties and biodiversity.

| | % Sand | % Silt | % Clay | %pH | % Pot | % Nit | % Oc | % OM | % Ca | % Mg | % Na | % Ex.Al | Animal index |
|---|---|---|---|---|---|---|---|---|---|---|---|---|---|
| % Sand | 1.000 | | | | | | | | | | | | |
| % Silt | −1.000 | 1.000 | | | | | | | | | | | |
| % Clay | 0.700 | −0.700 | 1.000 | | | | | | | | | | |
| % P$^H$ | −0.975 | 0.975 | −0.821 | 1.000 | | | | | | | | | |
| % Pot | −0.900 | −0.900 | −0.900 | 0.975 | 1.000 | | | | | | | | |
| % Nit | 0.462 | 0.462 | 0.564 | −0.605 | −0.718 | 1.000 | | | | | | | |
| % Oc | 0.300 | −0.300 | 0.500 | −0.462 | −0.600 | 0.975 | 1.000 | | | | | | |
| % OM | 0.300 | −0.300 | 0.500 | −0.462 | −0.600 | 0.975 | 1.000 | 1.000 | | | | | |
| % Ca | −0.900 | 0.900 | −0.500 | 0.872 | 0.800 | −0.616 | −0.500 | −0.500 | 1.000 | | | | |
| % Mg | −0.308 | 0.308 | −0.872 | 0.500 | 0.667 | −0.632 | −0.667 | −0.667 | 0.205 | 1.000 | | | |
| % Na | 0.900 | −0.900 | 0.500 | −0.872 | −0.800 | 0.616 | 0.500 | 0.500 | −1.000 | −0.205 | 1.000 | | |
| % Ex.Al | 0.700 | −0.700 | 0.200 | −0.616 | −0.500 | 0.359 | 0.300 | 0.300 | −0.900 | 0.051 | 0.900 | 1.000 | |
| Biodiversity index | 0.000 | 0.000 | 0.300 | −0.051 | 0.100 | −0.050 | −0.200 | −0.200 | −0.100 | 0.462 | 0.100 | −0.300 | 1.000 |

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
