# Peer review of "The Impact of Soil Erosion on Biodiversity Conservation in Isiala Ngwa North LGA, Southeastern Nigeria"

_sustainability, doi:10.3390/su11247192_

Round 1
Reviewer 1 Report
This work addresses the very important issue of maintaining tropical biodiversity under conditions of a strong anthropogenic pressure. The dependence of the quality and degree of soil erosion with the level of biodiversity shown by the authors is of some practical interest. The proposed recommendations will probably allow in the future to more successfully find a compromise between the interests of local residents and the conservation of biodiversity.
Author Response
AUTHORS’ REPLY TO REVIEWER 1.
TOPIC: Impact of soil erosion on biodiversity conservation in Isiala Ngwa North L.G.A. Abia State
There are no replies from authors to reviewer 1 because there is nothing that was reviewed by the reviewer 1. Reviewer 1 pointed out the relevance of the work to the conservation of biodiversity especially from the tropical area. In his words:
“ This work addresses the very important issue of maintaining tropical biodiversity under conditions of a strong anthropogenic pressure. The dependence of the quality and degree of soil erosion with the level of biodiversity shown by the authors is of some practical interest. The proposed recommendations will probably allow in the future to more successfully find a compromise between the interests of local residents and the conservation of biodiversity”,
the first reviewer made it clear that the work is ‘of some practical interest’ and should be encouraged.
Reviewer 2 Report
Dear Authors,
I revised the manuscript "Impact of soil erosion on biodiversity conservation in Isiala Ngwa North L.G.A. Abia State" submitted to the Sustainability Journal. The paper is interesting. However, I have some concerns, which need to be addressed before considering for final publication.
Check style and grammar in English. Some sentences are too long.
Use more references in the whole manuscript. Look for them in MDPI journals as well.
Section "Results" and "Discussion" are missing.
Explain some abbreviations e.g. "OC", "OM" etc.
References should be numbered in order of appearance and indicated by a numeral or numerals in square brackets, e.g., [1] or [2,3], or [4–6]. See instructions in the file "sustainability-template.dot".
Insert number of section.
Line 36. Change the phrase "at a phenomenal rate".
Line 46: Use more up-to-date data than 2010.
Line 95. Change to "Material and Methods".
Line 103-108. If possible also use Shannon Wiener diversity index, Pielou index, Simpson index or Brillouin index.
Line 109. See section 3.3 Formatting of Mathematical Components in the file "sustainability-template.dot".
Line 116-119. Where are Appendix 3 and Appendix 4?
Line 120. Enter the equation for Spearman’s rank correlation.
Line 134. “(-.0.671)” - is this correct?
Line 289. „qaudrats” - is this correct?
Author Response
AUTHORS’ REPLY TO REVIEWER 2.
Reviewer 2 made a lot of arguments and suggestions on where the work should be improved upon for better results. We have been able to tackle reviewer 2 comments as would be shown here.
Dear Authors,
I revised the manuscript "Impact of soil erosion on biodiversity conservation in Isiala Ngwa North L.G.A. Abia State" submitted to the Sustainability Journal. The paper is interesting. However, I have some concerns, which need to be addressed before considering for final publication.
Check style and grammar in English. Some sentences are too long.
The style of English has been improved upon. No more long sentences as complex sentences have been shortened to simple sentences for clearer understanding. As shown below:
Land is a resource for agricultural activities. Unregulated increase in land-use causes soil erosion and loss of biodiversity. Erosion is widely recognized as one of the main threats to soil [1]. Soil erosion is a challenge for sustainability of agriculture especially in tropical region. A critical global land degradation phenomenon affecting human beings is soil erosion. This is because humanity’s basic sources of livelihood are obtained from the land. FEIZNIA, and NOSRATI [2] CHAPPELL, et al [3] MOHAWESH, et al, [4] IPCC [5] stated that, land use changes worldwide have been recognized as capable of increasing the rate of soil erosion and loss of biodiversity. Intensive agricultural activities are reported to lead to soil erosion and loss of soil biodiversity [6] [7] [8] [9] [10].
Soil erosion results in the depletion of below ground biodiversity which include soil microorganisms. Vallejo, et al [11] and Gardi, et al [12] quoted in Lui, et al [13] stated that soils are one of the main living places of microorganisms and that microorganisms are involved in the decomposition of organic matter; play important roles such as the cycling and transformation of soil organic matter and soil nutrients which include carbon, nitrogen, phosphorus, and sulfur. These soil nutrients enhance agricultural productivity if they are not degraded by soil erosion. Worldwide, generally, soil is being degraded at a rapid rate. Globally, through soil erosion, about 2.8 tonnes of soil are lost per hectare annually t/(ha year) – [14]. Degraded soil is unproductive, which is determined by the degree of degradation to land damage. Centre for Science and Environment [15], stated that about 25-30 per cent of total cultivated land in India is affected by soil erosion. Also, Le ROUX, et al, [16] stated that in South Africa, over 70% of the nation’s land surface has negatively been impacted by varying levels and types of soil erosion. Similarly, FAO [17] indicates that without any conservation measures, the total area of rain-fed cropland in developing countries such as Latin America, Asia and Africa would get smaller by 544 million hectares in long-term because of soil erosion and land degradation. Loss of soil from agricultural land may cause environmental impacts as well as reducing soil productivity. LAL, et al., [18] stated that soil fertility, organic matter in the soil, plants rooting depth and plant-available water reserves can be decreased by soil erosion. The rates of soil erosion that exceed the generation of new topsoil area dynamic process may lead to a decline of soil productivity, and result in lower agricultural yield and income. The direct impact of soil degradation can be identified in the variations of agricultural output. However, this impact is affected by a number of socio-economic and agro-ecological variables. KUMAR and PANI, [19] state that soil’s physical degradation affects crop growth and yield by decreasing root depth, water availability and nutrient reserves. Thus, it leads to yield loss by affecting soil organic carbon, nitrogen, phosphorus, and potassium contents and soil pH. SCHERR [20] stated that, the effects of soil degradation vary with the initial soil conditions, types of soil, extent of soil degradation and crops. Increased food production is required by future world population [21] which is said to have grown to 7.06 billion in middle of 2012, after having crossed the 7 billion mark in 2011 [22]. Also, the 79.3 million people added to the overall global population each year has been consistent for nearly a decade [23]. This means that there is need to increase agricultural produce to feed this additional millions of people each year with food. Without good quality and nutrient rich soil, this is not possible. Hence damage, through soil erosion or in any other forms, to the soil is an indirect damage to agricultural production and ultimately food security. According to WALL, et al. [24], the implication of soil erosion extends beyond the removal of valuable topsoil. In fact, crop emergence, growth and yield are directly affected through the loss of natural nutrients. BATHRELLOS, et al., [25], stated that the main on-site impact of soil erosion is the reduction of soil quality which results from the loss of the nutrient-rich upper layers of the soil. The erosion of soil is one of several natural and human threats to sustained soil productivity, which may become irreversible if not mitigated [18]. It threatens man’s source of food, livelihood and destroys man’s property and investments [26].
Thus,…………..
Use more references in the whole manuscript. Look for them in MDPI journals as well.
More references have been included in the manuscript and a work from the Sustainability journal which is an MDPI JOURNAL was also cited in the manuscript.
Section "Results" and "Discussion" are missing.
RESULTS and DISCUSSION section has now been added to the work.
Explain some abbreviations e.g. "OC", "OM" etc.
The abbreviations “OC” Organic Carbon and “OM” Organic Matter are corrected and included in the first time they appear in the manuscript while the abbreviation like OC and OM were used where they appear subsequently in the manuscript
References should be numbered in order of appearance and indicated by a numeral or numerals in square brackets, e.g., [1] or [2,3], or [4–6]. See instructions in the file "sustainability-template.dot".
References have been numbered in order of appearance using the style of Sustainability journal.
Insert number of section.
Number of section have been inserted.
Line 36. Change the phrase "at a phenomenal rate".
The entire section has been rewritten and so there are so many thing improvements on the section
Line 46: Use more up-to-date data than 2010.
Same as Line 36
Line 95. Change to "Material and Methods".
It has been done
Line 103-108. If possible also use Shannon Wiener diversity index, Pielou index, Simpson index or Brillouin index.
Shannon Wiener diversity index was actually used in the work
Line 109. See section 3.3 Formatting of Mathematical Components in the file "sustainability-template.dot".
It has been done. Thank you
Line 116-119. Where are Appendix 3 and Appendix 4?
They have been changed to appendix 1 and 2 and attached.
Line 120. Enter the equation for Spearman’s rank correlation.
The Formula for Spearman Rank Correlation
rR = 1- ---------------------------- (2)
where n is the number of data points of the two variables and di is the difference in the ranks of the ith element of each random variable considered. The Spearman correlation coefficient, ρ, can take values from +1 to -1.
A ρ of +1 indicates a perfect association of ranks A ρ of zero indicates no association between ranks and ρ of -1 indicates a perfect negative association of ranks.The closer ρ is to zero, the weaker the association between the ranks.
Line 134. “(-.0.671)” - is this correct?
It has been corrected to [ -0.671]
Line 289. „qaudrats” - is this correct?
No. It has also been corrected to quadrats. Thank you.

Reviewer 3 Report
Some changes I think must be done:
l. 33: "...is from the land." Better: ... is obtained from the land. Table 3: Plant Index would not be Animal Index? Table 5: Plant Index would not be Biodiversity Index?I thought that, in the PCA analyze, when there are, for instance, negative variable indexes and a negative plant index, the relationship must be direct, but sometimes the conclusions are not in accordance with this.
Author Response
AUTHORS’ REPLY TO REVIEWER 3.
Reviewer 3 made some corrections that very vital to the improvement of the work. We were able to give a reply to the works as shown below:
REVIEWER 3
Some changes I think must be done:
33: "...is from the land." Better: ... is obtained from the land.
This comment has been modified in the manuscript by a complete revision of the manuscript. The introductory section has been improved upon tremendously. Thank you.
Table 3: Plant Index would not be Animal Index? Table 5: Plant Index would not be Biodiversity Index?
The corrections pointed out have effected from Tables 3 and 5. Both plant index and animal index have all been correctly placed in their rightful positions within the tables. Thank you.
I thought that, in the PCA analyze, when there are, for instance, negative variable indexes and a negative plant index, the relationship must be direct, but sometimes the conclusions are not in accordance with this.
Yes, you are very correct. The relationship is direct either in the positive or negative direction depending on the strength of the figure under consideration which must be from + or - 0.5 and above. If it has a negative sign, the direct relationship should be negative while the positive sign indicates a direct positive relationship. Thank you.

Round 2
Reviewer 2 Report
From my point of view, the mistakes were corrected and the paper was overall improved. I recommend to accept it in present form.